

# Automated identification and segmentation of urine spots based on deep-learning

Xin Fan[1], Jun Li[2] and Junan Yan[3]

[1] Medical School, Guangxi University, Nanning, Guangxi, China
[2] School of Physical Science and Technology, Guangxi University, Nanning, Guangxi, China
[3] Naval Medical Center, Naval Medical University, Shanghai, Shanghai, China

## ABSTRACT

Micturition serves an essential physiological function that allows the body to eliminate metabolic wastes and maintain water-electrolyte balance. The urine spot assay (VSA), as a simple and economical assay, has been widely used in the study of micturition behavior in rodents. However, the traditional VSA method relies on manual judgment, introduces subjective errors, faces difficulty in obtaining appearance time of each urine spot, and struggles with quantitative analysis of overlapping spots. To address these challenges, we developed a deep learning-based approach for the automatic identification and segmentation of urine spots. Our system employs a target detection network to efficiently detect each urine spot and utilizes an instance segmentation network to achieve precise segmentation of overlapping urine spots. Compared with the traditional VSA method, our system achieves automated detection of urine spot area of micturition in rodents, greatly reducing subjective errors. It accurately determines the urination time of each spot and effectively quantifies the overlapping spots. This study enables high-throughput and precise urine spot detection, providing important technical support for the analysis of urination behavior and the study of the neural mechanism underlying urination.

## INTRODUCTION

Micturition, a complex neuromuscular activity, is governed by the central nervous system and is vital for maintaining physiological health (*Adriaansen et al., 2017*; *Yao et al., 2018*). In many mammals, this basic physiological process also fulfills several biological functions, including socialization, reproduction, and territorial marking (*Hou et al., 2016*; *Keller et al., 2018*). Dysfunctions in the lower urinary tract, such as urinary frequency, urgency, and incontinence, can significantly impair a patient's quality of life (*Abarbanel et al., 2003*). Consequently, a comprehensive understanding of neuromodulatory mechanisms during both normal and abnormal micturition is essential for guiding clinical diagnostics and treatments (*Yu et al., 2014*). Various methods exist for assessing lower urinary tract function, ranging from noninvasive voiding spot assay (VSA) and metabolic cage testing to more invasive urodynamic studies (*Sartori, Kessler & Schwab, 2021*).

Corresponding author
Junan Yan, junan_yan@aliyun.com

In experimental animal models, particularly rodents, VSA techniques are widely used by many researchers to evaluate spontaneous micturition patterns, thus playing a pivotal role in exploring lower urinary tract physiology and its neurobiological underpinnings. VSA was validated for evaluation of bladder function and found correlations with certain cystometric pressures but not others (*Hodges et al., 2008*). Some researchers associated ketamine treatment in mice with increased voiding on VSA despite unchanged cystometric results (*Rajandram et al., 2016*). The voiding frequency in Htr3a mutant mice was evaluated through VSA analysis, which was not corroborated by cystometric assessments (*Ritter et al., 2017*). The VSA approach was also utilized for sex-specific behavioral analysis related to sex hormones (*Wu et al., 2009*). In addition, some studies were conducted to assess age-related urothelial changes in male mice with a 4-hour free-access assay (*Desjardins, Maruniak & Bronson, 1973*). A 4-hour water-restricted assay was used by associating 1 integrin knockout in mice with mechanosensory bladder overactivity (*Kanasaki et al., 2013*). However, within the previous studies, traditional VSA techniques present challenges in analysis, especially when dealing with overlapping urinary spots, which are time-consuming and subject to data analysis variability and errors. Additionally, these methods often fail to precisely determine the timing of micturition behaviors (*Wegner et al., 2018*). To overcome these limitations, the development of an efficient, automated VSA analysis method is imperative. Such a method would enhance data reproducibility and comparability, enable unbiased urine spot detection and segmentation, and reduce the time and potential errors associated with manual analysis. Ultimately, this advancement would offer more accurate insights into neural control mechanisms and micturition behavior interpretation.

Deep learning, a significant branch of artificial intelligence (AI), has achieved remarkable achievements in image processing. Its applications span from object detection, classification, and localization to behavior recognition, action tracking, and event detection, becoming increasingly prevalent (*Abdollahzadeh et al., 2021*; *Hillsley, Santos & Rosales, 2021*). In medical image analysis, deep learning has successfully identified fractures in X-ray images and detected tumors in MRI scans (*Black et al., 2020*; *Wang et al., 2023*). These advancements highlight AI's potential in interpreting visual information and offer novel perspectives and tools for assessing lower urinary tract function. Deep learning's real-time processing capabilities are particularly beneficial for dynamic bioprocess analysis, including automated urine spot video evaluations.

This article introduces an innovative deep learning-based method for the automatic identification and segmentation of urine spots. Utilizing advanced algorithms and consistent image quantization techniques, this method completely automates urine spot analysis. It effectively recognizes and segments crucial features in urine spot images, enhancing the speed, accuracy, and user-friendliness of assessing lower urinary tract dysfunctions. Experimental results demonstrate that this method significantly reduces subjective errors and processes large volumes of image data more efficiently and accurately than traditional VSA techniques. This results in considerable reductions in analysis time and cost while improving the accuracy and efficiency of urine phenotyping in mice. Offering a more objective and standardized assessment, it provides reliable data for studying lower urinary tract dysfunctions in mice. Furthermore, this study is the first to

integrate video recording with deep learning for continuous observation and quantitative analysis of urination behavior in mice. It can continuously capture the temporal and spatial distribution and frequency of urination events, as well as their correlation with mice behavior. This innovative method offers substantial technical support for analyzing urination mechanisms, screening therapeutic drugs, and developing new high-throughput urinary function assessment tools.

## MATERIALS & METHODS

### Animal

All animal procedures were strictly conducted according to institutional guidelines and protocols, having obtained approval from the Institutional Animal Care and Use Committee of the Guangxi University (GXU-2024-003). Four male and female mice, aged between 2 and 5 months with a C57BL/6J genetic background were obtained from the Laboratory Animal Center at the Guangxi University. Subsequently, the mice were housed in groups, with a maximum of five individuals per cage, under a 12-hour light-dark cycle while having ad libitum access to water and food (*Keller et al., 2018*). Euthanasia of the animals was carried out using an intraperitoneal injection of a dosage three times higher than that of pentobarbital sodium before finishing the experiments. This method was employed to ensure humane and ethical treatment of the mice during the euthanasia process.

### Spontaneous void spot assay

In this experiment, a double-layer structure device was designed to observe the urination behavior of mice. The size of the upper structure was 27 × 15 × 20 cm. The lower compartment was equipped with a 365 nm ultraviolet (UV) lamp (DJ Black-24BLB; ADJ Lighting, Los Angeles, CA, USA). Neutral filter paper (Xinxing) absorbed the urine of the mice, and the lower compartment had mirrored walls. The UV lamp was activated and videos were captured using a Lenovo computer equipped with Logitech software and a video camera with 1,280 × 720 pixel resolution. Under these conditions, urine spots deposited on the filter paper could be detected by UV illumination. The experimental animals were domesticated in the behavioral chamber for three consecutive days before the experiment (Fig. 1A) (*Verstegen et al., 2019*).

### Image acquisition and analysis

After the experiment was completed, the recording device was turned off and a video describing the urine diffusion process was generated. The filter paper at the bottom was air-dried and returned to its original position, while the overhead infrared light was turned off. UV light was utilized to enhance the visibility of the urine spots, and images were captured with a camera. Throughout the process, we ensured that the pixel dimensions of both images and videos were 1,280 × 720, and collected mouse urination videos as training samples (Fig. 1B). The boundary information of each urination point was manually labeled (Fig. 2A). Next, we used our customized AutoVSA software for analysis. Figure 1 shows an example of filter paper under UV irradiation. Filter paper photos were saved in JPEG format.

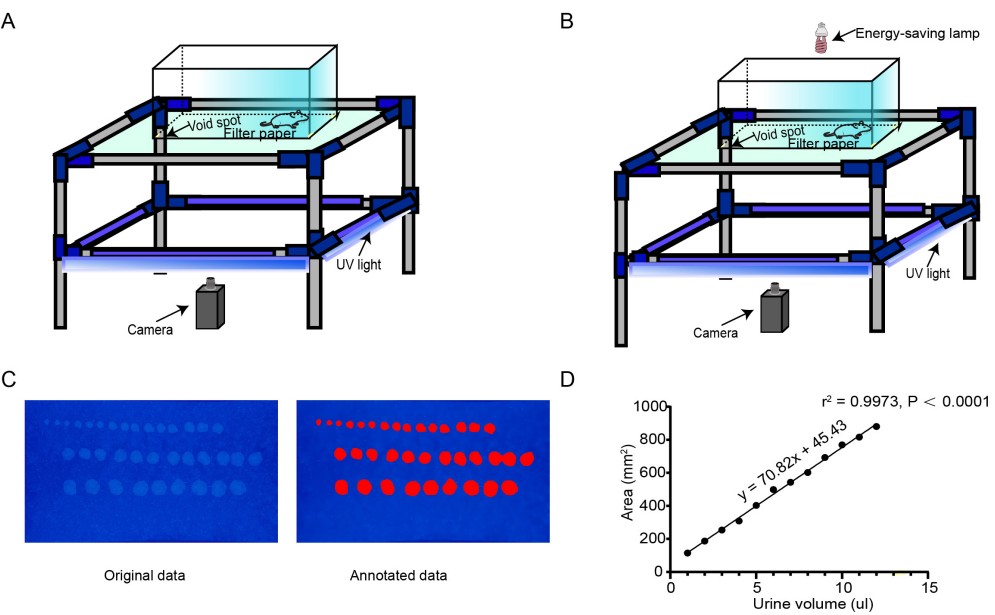

**Figure 1** **AutoVSA recording chamber device and urinary spot quantification analysis.** (A) Spontaneous void spot assay photographic device. (B) Real-time video recording and storage device; schematic representation of the void-spot assay for lower urinary tract assessments in urology. (C) Representative images of filter paper with known urine volumes. (D) Calibration curve (subfigures can be accessed *via* https://doi.org/10.6084/m9.figshare.25539451.v1).

## Conversion from area to volume

Mouse urine was used as an intermediate medium to enable quantitative determination of urine volume. Different volumes of urine were pipetted onto experimental neutral filter paper, and the spot area was converted to volume by constructing a standard curve correlating the known urine volume to the corresponding urine spot area on the filter paper (Fig. 1C) (*Sugino et al., 2008*). When the volume was below 100 microliters, there was a linear relationship between volume and area with an $R^2$ value of 0.9973 (Fig. 1D). Thus, urine volume can be accurately calculated based on different filter paper types and urine spot sizes.

## Statistical analysis

Statistical analysis was performed by comparing the two independent groups. For normally distributed data, the Student's $t$-test was used. In cases where the data deviated from a normal distribution, a nonparametric test for independent samples was used. $p$-values below 0.05 were considered statistically significant differences.

## AutoVSA

AutoVSA is a deep learning-based approach for automatically segmenting VSA images and videos. It comprises a detection network and a tracking network to perform instance segmentation and location tracking of urine spots, utilizing the YOLACT (*Bolya et al., 2019*) and DeepSort (*Pujara & Bhamare, 2022*) frameworks.

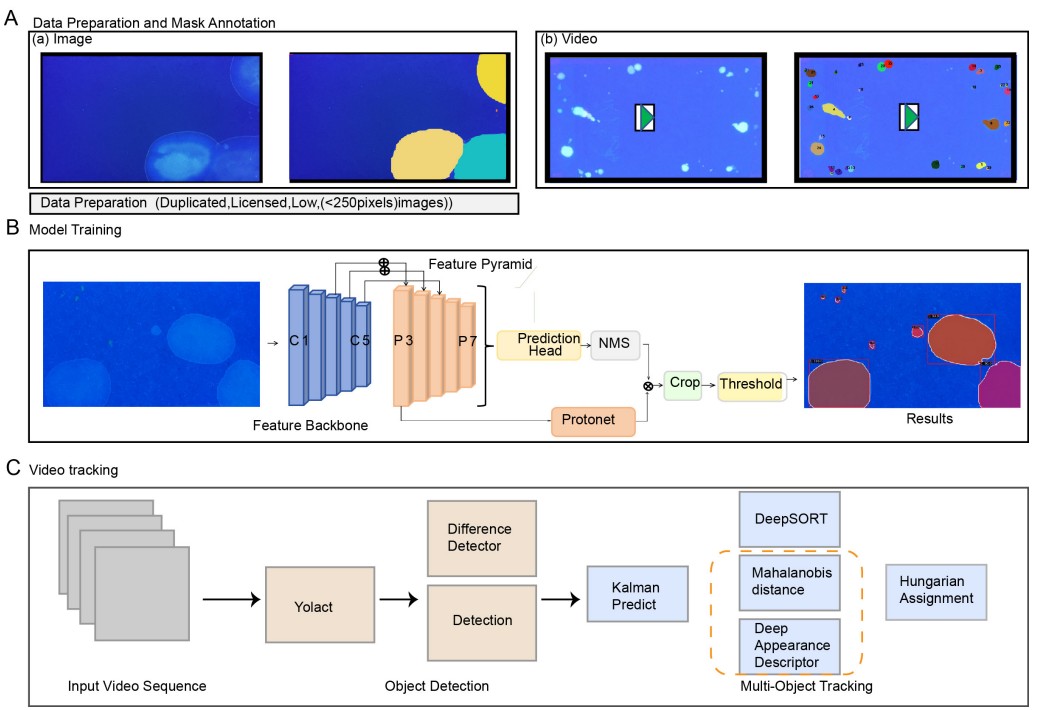

**Figure 2   Workflow for processing sample urinary spots Data collection, annotation, and method framework pipeline.** (A) Data collection and annotation. (B) Spot detection network structure, including computer-generated cross-sectional crops of the spot's surface, averaging classification scores, and visual comparisons of selected results on the training dataset. (C) AutoVSA tracking framework.

## Detection framework

In this study, the YOLACT algorithm is employed for the spot detection. The algorithm integrates rapid localization with precise pixel-level segmentation techniques. YOLACT uses Feature Pyramid Networks (FPNs) and Depth Separable Convolution for feature extraction. An anchor frame mechanism predicts the target's location and species. The second stage involves a series of prototypes and a novel linear combination of coefficients, significantly speeding up the segmentation process by dynamically generating segmentation masks for each instance and predicting mask coefficients concurrently (Fig. 2B).

## Tracking framework

The tracking framework primarily relies on the DeepSort algorithm. This algorithm encompasses state estimation, trajectory processing, and matching challenges. It leverages target motion and appearance data to reduce object ID switches. The Kalman filter predicts the target position and is integrated with the Hungarian algorithm and Intersection of Union (IoU) for trajectory estimation. Time and position data are recorded to track urination points in videos (*Yadav et al., 2022*). The marsupial distance quantifies motion information correlation, reflecting uncertainty in state measurements (*Durve et al., 2023*; *Veeramani, Raymond & Chanda, 2018*). Given the relatively slow diffusion rate of urine

spots, emphasizing motion information correlation as the primary matching metric is instrumental in analyzing the dynamics of the urine spots (Fig. 2C).

## Training of AutoVSA

Due to the limited data, image enhancement techniques like translation, rotation, and affine transformation were applied during training to produce three derivative images from each original image. A total of 2,600 images were utilized for the training and analysis of the deep learning model. All images have a uniform resolution of 1280*720 pixels .The LabelMe image annotation tool generated a JSON format file, segmenting the image dataset into a 9:1 training-to-validation ratio. The pre-trained YOLACT model and PyTorch framework were selected for parameter tuning, employing a stochastic gradient descent (SGD) optimization algorithm with a learning rate of 0.001, momentum of 0.9, and weight decay of 0.0001. Post-training, the model was applied to urine spot images and validated against manually labeled data (*Guan et al., 2017*; *Xu et al., 2023*). Leveraging the robust performance of ResNet-50 in image classification, YOLACT was chosen to achieve accurate pixel-level segmentation.

## Postprocessing

Watershed algorithms have proven effective in manually analyzing overlapping urine spots, particularly in quantifying and resolving overlaps. We utilized these algorithms for automated segmentation to address the separation of overlapping mouse urine spots. The method applies a watershed algorithm to automatically segment the target region's gray gradient by mimicking water flow, eroding filled edges, locating the center of mass, and reconstructing the boundary. This automated process effectively distinguishes non-concentric overlapping circles into distinct regions, accurately representing individual mouse urine spots (Fig. 3).

## Evaluation and statistical analysis

To ensure the accuracy of the results, this study conducted experimental evaluations on a standard test dataset that is widely recognized as a benchmark for rigorously testing the capabilities of instance segmentation methods. The raw image data collected were organized into the standard COCO dataset format after data preprocessing of the images. In these experiments, this article uses commonly used metrics such as AP (average precision) and the mean average precision (mAP) to evaluate the accuracy of the detection algorithms (*Ran et al., 2023*). mAP, as a commonly used evaluation metric in the field of computer vision, is widely used in tasks such as instance segmentation, object detection, and image categorization, and can provide a more comprehensive evaluation result (*Cattaneo et al., 2020*; *Sanchez, Romero & Morales, 2020*).

The study employs a suite of metrics—Precision (P), Recall (R), F1 score, and mean, average precision (mAP)—to benchmark the efficacy of surgical tool detection algorithms, see [22] for more details on these parameters.

$$P(\%) = \frac{TP}{TP + FP} \times 100$$

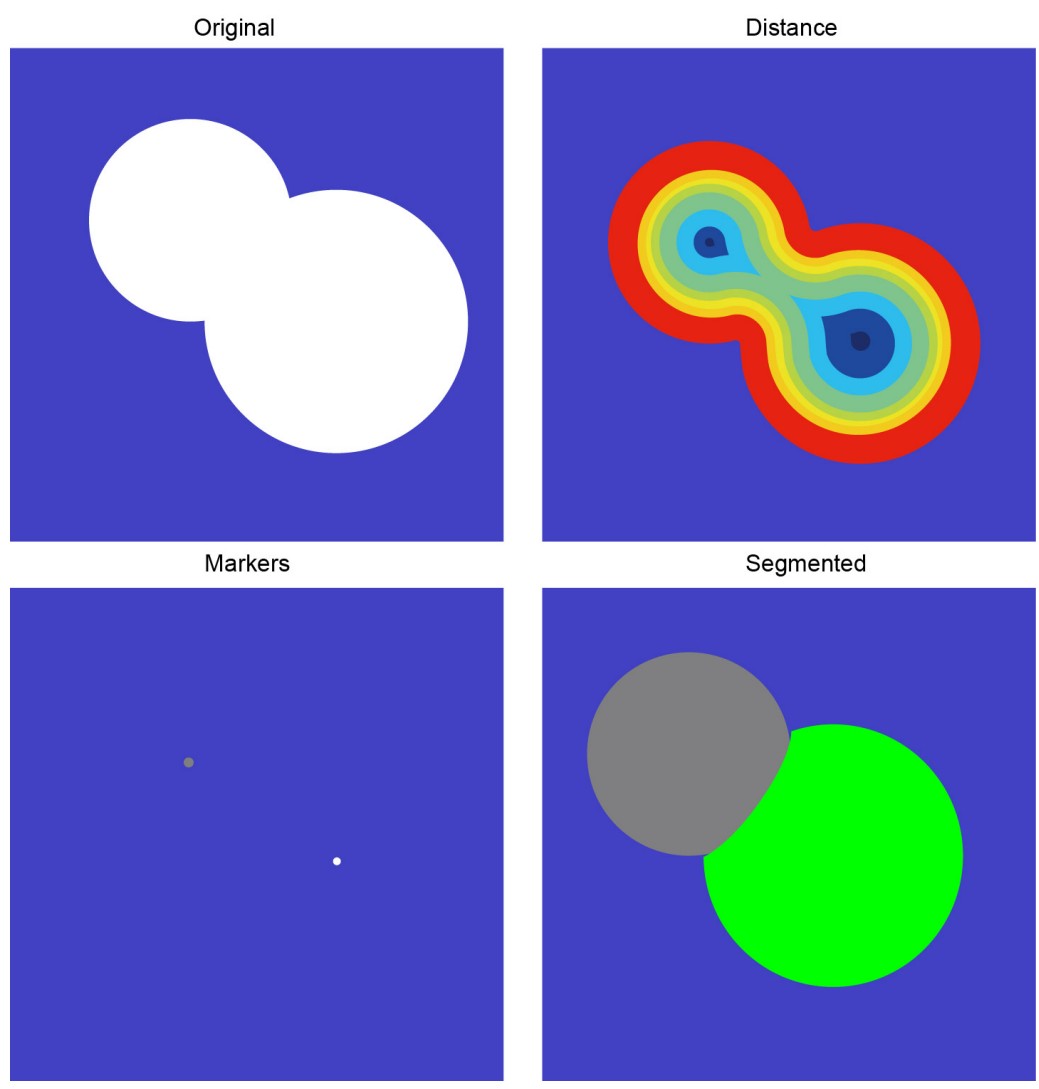

**Figure 3** Overlapping urine spot segmentation process.

$$R(\%) = \frac{TP}{TP + FN} \times 100$$

$$F1(\%) = \frac{2PR}{P + R}$$

$$AP = \int_0^1 PdR$$

$$mAP = \frac{1}{n}\sum_{i=1}^{n} AP_i.$$

Precision quantifies the ratio of accurately identified positives within the set of predicted positives, while Recall assesses the proportion of true positives out of all actual positives. In this context, true positives (TP) denote correctly identified positive instances, false positives (FP) denote incorrectly labeled positives, true negatives (TN) refer to correctly labeled negatives, and false negatives (FN) indicate wrongly labeled negatives. The computation of P, R, and F1 follows established formulae. Importantly, mAP serves as a metric for evaluating object detection accuracy, averaging the AP across categories, derived from the area under the Precision-Recall (PR) curve. Here, 'n' represents the category count. The research adopts prevalent COCO challenge metrics for object detection assessment, including AP and its variants $AP_{50}$, $AP_{75}$, $AP_S$ (small object area), $AP_M$ (medium object area), and $AP_L$ (large object area), with $AP_{50}$ and $AP_{75}$ denoting AP at IoU thresholds of 0.5 and 0.75, respectively. The metrics $AP_S$, $AP_M$, and $AP_L$ denote the Average Precision (AP) for object detections of varying scales, with $AP_S$ applying to bounding boxes under $32^2$ pixels, $AP_M$ to those within $32^2$ to $96^2$ pixels, and $AP_L$ to those exceeding $96^2$ pixels. Given the dataset's exclusive inclusion of large surgical instruments, characterized by bounding boxes surpassing $96^2$ pixels, computation of $AP_S$ and $AP_M$ is rendered irrelevant. Moreover, within this dataset, the $AP_L$ metric is observed to converge with the overall AP measure. $AP_{50}$ represents the average accuracy when the IoU threshold is set to 50%. If the IoU between the predicted bounding box and the true bounding box is greater than or equal to 50%, it is considered to be a correct detection. $AP_{50}$ is more lenient, which reflects the model's ability to detect larger-sized speckles or targets with more distinctive boundaries. $AP_{75}$ denotes the average accuracy when the IoU threshold is set to 75%, which is a more stringent evaluation criterion. $AP_{75}$ can be a more precise measure of the model's detection accuracy for small-sized spots or targets with less distinct boundaries.

## RESULTS

AutoVSA represents an efficient deep learning-based technique for the automatic detection and segmentation of urine spots. To evaluate its accuracy and efficiency in analyzing autonomously voided urine spot data, the performance of AutoVSA was assessed in comparison with state-of-the-art tools like ImageJ and SOLOv2. Additionally, AutoVSA's predictions were contrasted with manually labeled ground truth and alternative computational methods. The comparative results are illustrated in Fig. 4. As depicted, AutoVSA demonstrates high accuracy in detecting and segmenting urine spots across various image dataset sizes, exhibiting a low omission rate in identification and providing precise edge outlining for urine spots of varying sizes in typical images (Fig. 4).

Algorithm yields superior segmentation results. This study tested its effectiveness on urine spot segmentation of various sizes, as presented in Fig. 4. To evaluate segmentation accuracy, we employed mAP metric across different scales. A higher mAP value indicates closer alignment of the algorithm's segmentation with manually labeled results. The results show that the algorithm used in this article has good robustness and accuracy in segmenting objects. To evaluate the performance of the model at the pixel level, this article employs the mAP evaluation metric at different scales (*Liu et al., 2020*). Of particular note, the YOLACT

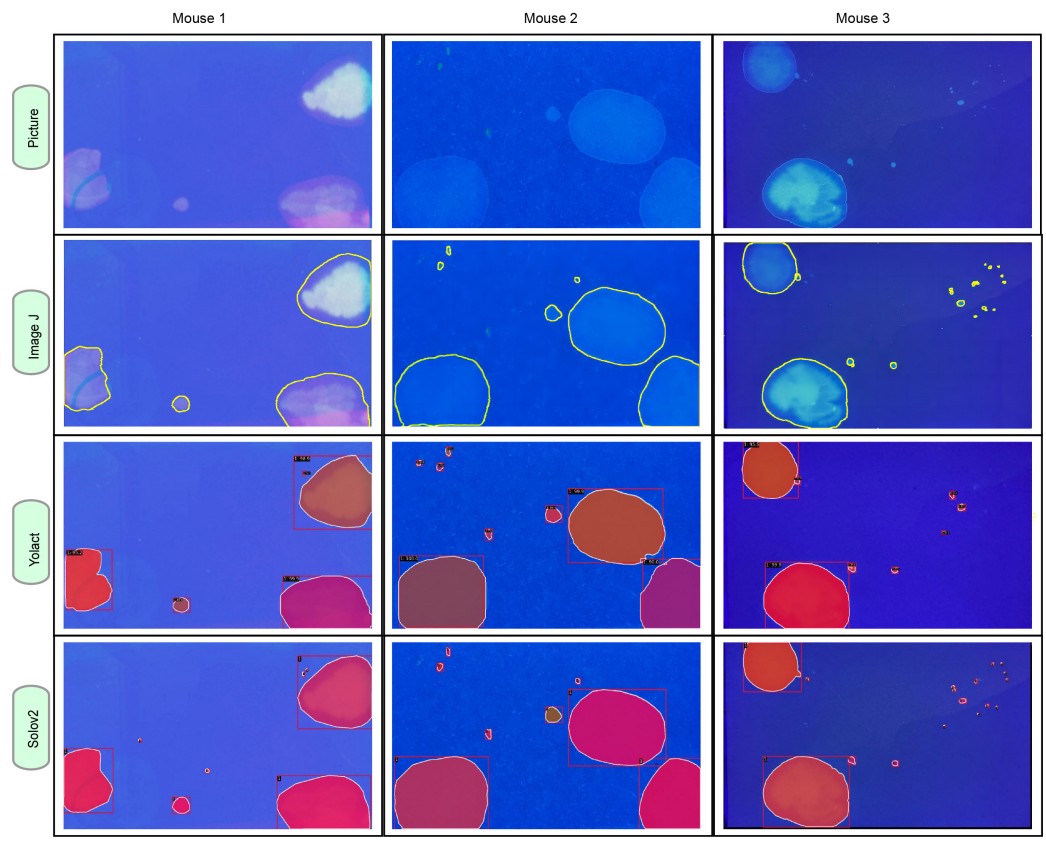

**Figure 4    Instance segmentation results of urine spot images on C57BL/6J mice.** The segmentation results from AutoVSA, Image J of urine spots are compared with the region of interest (ROI).

**Table 1    Comparison of pixel-level segmentation accuracy on the dataset.**

|  | Backbone | AP(%) | $AP_{50}$(%) | $AP_{75}$(%) | $AP_S$(%) | $AP_M$(%) | $AP_L$(%) |
|---|---|---|---|---|---|---|---|
| YOLACT | Res-50-FPN | 29.3 | 56.1 | 27.2 | 18.7 | 59.8 | 73.6 |
| SOLOv2 | Res-50-FPN | 22.3 | 36.7 | 24 | 9.9 | 57.8 | 76 |

algorithm achieves excellent performance in detection and segmentation at small (18.7%), medium (59.8%), and large (73.6%) scales as presented in Table 1. These findings, detailed in Fig. 4 and Table 1, highlight its exceptional performance. However, some limitations were noted in SOLOv2, including instances of incomplete segmentation and discrepancies in segmentation size (Fig. 4).

## Comparison with different architectures & Fiji

Comparative analyses using Fiji and AutoVSA showed that the points on the scatterplot were in close agreement with the equation line, indicating that there were no significant differences between the mice and the corresponding controls in terms of number, area, and frequency across the 30 images (Fig. 5).

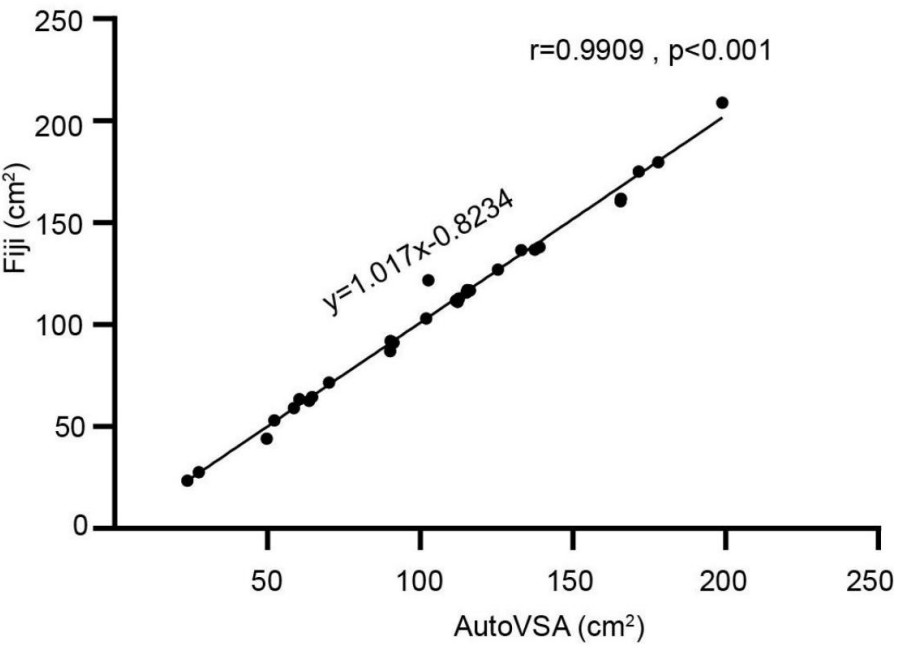

**Figure 5  Comparison between Fiji and our software.** Exemplary images of filter paper analysis obtained through Python and Fiji software.

Furthermore, this study implemented dynamic detection and segmentation of urine spots in video data through a GUI interface using AutoVSA (Fig. 6). The method has the advantage to accurately detect and segment detailed information for each urination event, including spot number, frequency, interval, and location (Fig. 7). In summary, AutoVSA demonstrates robust performance on both image and video data at various scales, offering significant advantages over other tools. This evidence supports the method's feasibility and validity in assessing the volume of tiny cavities in normal mice (Table 2).

The results indicate that AutoVSA offers higher accuracy and reliability in urine spot detection and segmentation tasks compared to the traditional ImageJ method. It effectively recognizes and segments urine spots in various images and excels across different image scales, significantly outperforming existing tools. The study confirms AutoVSA's feasibility and effectiveness in assessing urinary output in mice, providing a precise and efficient tool for studying lower urinary tract function.

## DISCUSSION

In recent years, the VSA method has been widely used in the field of spontaneous micturition research due to its simplicity, low cost, and non-invasiveness (*Chen et al., 2017*; *Sugino et al., 2008*). The quantitative analysis of VSA is indispensable for an in-depth understanding of the normal physiological and pathological states of the urinary system; however, it faces the challenges of the complexity of the traditional manual urological spot analysis methods (*e.g.*, ImageJ) (*Carattino et al., 2023*), which is time-consuming and susceptible to the

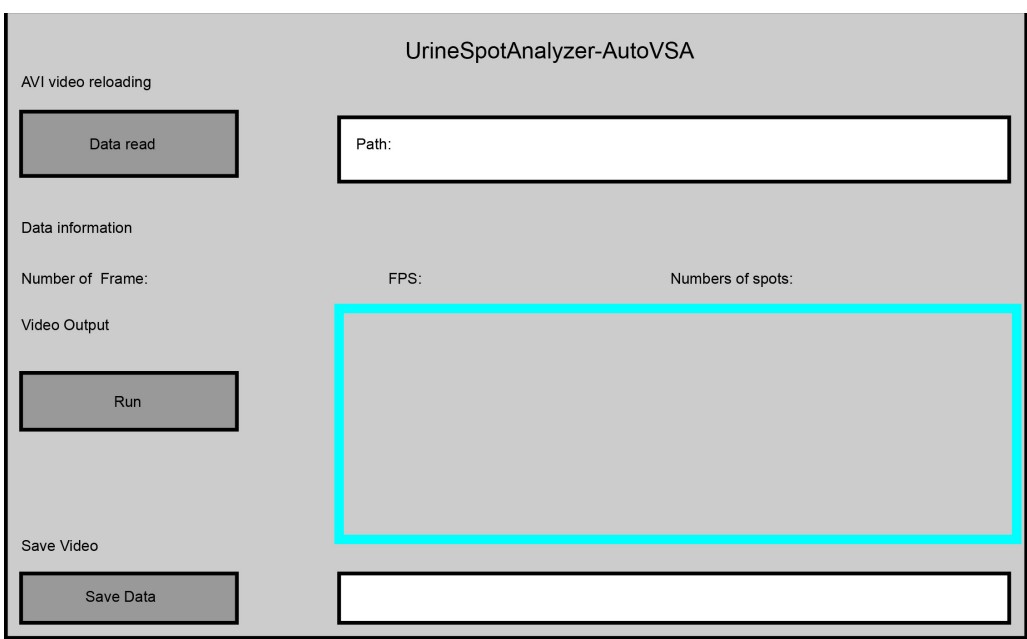

**Figure 6  GUI interface of the software for quantitative analysis of video data with urine spots. Capture images show the tracking effect of video with Urine spots in four time periods.**

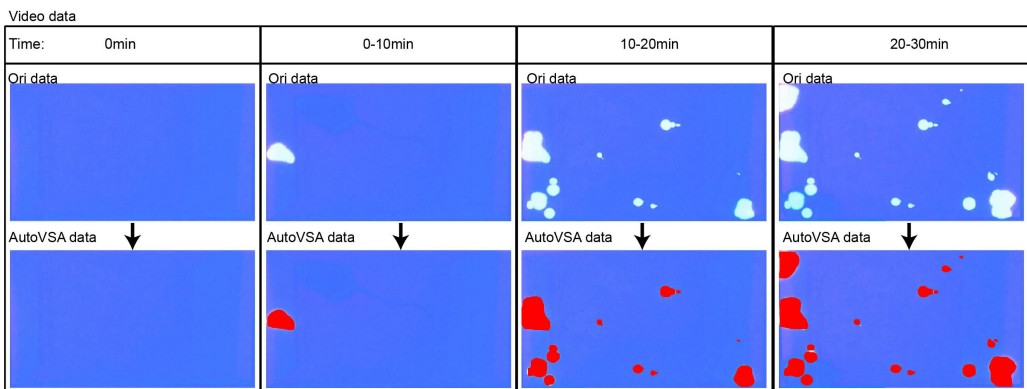

**Figure 7  Quantitative analysis of video data with urine spots.** Capture images show the tracking effect of video with Urine spots in four time periods.

influence of subjective judgment, especially in the scenario of overlapping spots (*Chen et al., 2017*; *Hill et al., 2018*; *Wegner et al., 2018*). This is especially true in the case of overlapping urine spots. To overcome these challenges, this study introduces AutoVSA, an automated image processing method employing convolutional neural networks for the automatic detection and segmentation of urine spots. This enhances data processing efficiency and accuracy. AutoVSA allows researchers to assess murine urination function more efficiently and non-invasively, automatically quantifying key parameters like frequency, volume, and distribution. Additionally, this study pioneers the integration of video-based dynamic

**Table 2** The results from the laboratory-standard VSA analyses (Fiji) were compared with those of AutoVSA analyses.

| Analytical method | Average number of spots counted | Average area (cm$^2$) | Total of time |
|---|---|---|---|
| Fiji | $19 \pm 2.72$ | $105.33 \pm 8.40$ | 40 min |
| SOLOv2 | $17 \pm 2.26$ | $194.98 \pm 8.90$ | 2s |
| AutoVSA | $18 \pm 2.21$ | $104.45 \pm 8.40$ | 2s |

**Notes.**
Values are means $\pm$ SE; $n = 30$.

analysis in VSA technology, enabling precise differentiation of voiding events over various time scales. This innovation marks a significant advancement in VSA technology.

In the realm of medical image segmentation, instance segmentation techniques are extensively applied in biomedical research, including cell image segmentation and lung tumor detection (*Black et al., 2020*; *Dai et al., 2023*; *de Brevern, Jia & Sun, 2017*; *Wang et al., 2023*). This article assesses the performance of YOLACT and ImageJ in detecting and segmenting urine spots in mice. The findings reveal that both methods closely align with manual analysis in measuring urine spot number and area. YOLACT, a single-stage, end-to-end instance segmentation network, surpasses traditional methods by integrating a feature pyramid network (FPN) for multi-scale semantic information fusion and fine-grained feature extraction. Data in Table 1 underscores YOLACT's leading edge in segmenting urine spots with notable precision and comprehensive coverage, reflected in mAP$_S$ (18.7%), mAP$_M$ (59.8%), and mAP$_L$ (73.6%) across different scales. Cross-validation with Fiji and AutoVSA aligns closely with theoretical expectations, demonstrating YOLACT's reliable quantification of urination metrics without significant deviation from the control group in the evaluated image set (Fig. 5). It excels in segmenting urine spots of varying sizes and significantly enhances processing speed and efficiency. We find that while the deep learning model fails to reach the level of human experts in some cases, it shows similar or even higher performance in most cases. Errors in the model can affect the analysis results in two ways: false negatives through missed urine spotting, and false positives through misidentifying other objects as urine spots. Missed detections can lead to an underestimation of voiding events, while false positives can lead to an overestimation of voiding events. Our model shows advantages in reducing leakage detection, which is particularly important for those studies that investigate parameters such as frequency and volume of urination and studies of neural mechanisms related to urination. This study pairs YOLACT with the DeepSort target tracking algorithm, resulting in efficient and accurate detection and segmentation of urine spot video, greatly outperforming traditional manual methods. This automated urine spot analysis system will aid researchers in understanding the physiological mechanisms of urination, potentially accelerating drug screening and the development of new therapies. Overall, this approach facilitates high-throughput, unbiased analysis of urinary biomarkers, serving as a valuable tool in urinary function studies. Comparison with other models, like YOLOv7, would be performed in the future for a further investigation.

In this research, we have rigorously explored and investigated the challenges faced by VSA image processing, and in particular, we have proposed an innovative solution

to the uraemic overlap problem (*Luo et al., 2023*). Researchers typically employ various algorithms, including region-based, edge, thresholding, and cluster analysis. The watershed segmentation algorithm, a region-based method, is recognized for its straightforward implementation and effective contour extraction. However, it is prone to noise sensitivity and image interference, leading to over-segmentation. To mitigate this issue, we introduce a novel algorithm for color image segmentation that combines deep learning with the watershed technique. Initially, advanced filtering techniques preprocess the image to reduce noise impact. Then, a deep learning model identifies the urethra, and segmentation is executed using the watershed algorithm on gradient images. This approach curtails over-segmentation caused by minor variations by setting an appropriate threshold, yielding more precise and reliable automatic segmentation results.

With the ongoing advancement of deep learning technology, we anticipate further enhancements in various aspects, such as diversifying network architectures, innovating loss functions, and optimizing training strategies. To advance the method's performance, we aim to overcome the current limitations and investigate new techniques to further refine urine spot detection in the future. We will enhance model performance by employing larger, more diverse datasets and broadly implementing automatic identification and segmentation for rapid online detection and localization of task-specific null spots. The study demonstrates that deep learning can effectively automate VSA image analysis for recording and identifying urination patterns, enabling precise, high-throughput detection and supporting research into urination behavior and its neural mechanisms.

## CONCLUSIONS

This study investigated the detection and segmentation of mouse urine spots, proposing a novel automated analysis method based on deep learning. This method employs deep learning algorithms for the automatic identification and segmentation of urine spots, effectively handling overlapping spots through post-processing with the watershed algorithm. Compared to traditional manual processing, this study achieves standardization and automation of the entire analysis process, minimizing user intervention. Experimental results indicate superior performance over traditional methods in segmenting urine spots under ultraviolet irradiation. This automated analysis method facilitates standardization in urine spot analysis, advances lower urinary tract function studies, and enhances understanding of the physiological mechanisms of the urinary system. In conclusion, this study presents a new, feasible approach for the automatic detection and segmentation of urine spots in mice using deep learning techniques.

### Funding

The research was funded completely by a grant from the National Natural Science Foundation of China (Grant/Award Number: 31970946). The funders had no role in

study design, data collection and analysis, decision to publish, or preparation of the manuscript.

## Grant Disclosures

The following grant information was disclosed by the authors:
The National Natural Science Foundation of China: 31970946.

## Competing Interests

The authors declare there are no competing interests.

## Author Contributions

- Xin Fan conceived and designed the experiments, performed the experiments, analyzed the data, prepared figures and/or tables, authored or reviewed drafts of the article, and approved the final draft.
- Jun Li performed the experiments, analyzed the data, authored or reviewed drafts of the article, and approved the final draft.
- Junan Yan conceived and designed the experiments, authored or reviewed drafts of the article, and approved the final draft.

## Animal Ethics

The following information was supplied relating to ethical approvals (i.e., approving body and any reference numbers):
Guangxi University provided full approval for this research (GXU-2024-003).

## Data Availability

The data and code are available at figshare: Fan, Xin (2024). mmdetection.zip. figshare. Dataset. https://doi.org/10.6084/m9.figshare.25538596.v1.

## Supplemental Information

Supplemental information for this article can be found online at http://dx.doi.org/10.7717/peerj.17398#supplemental-information.

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
