# Peer review of "Automated identification and segmentation of urine spots based on deep-learning"

_PeerJ, doi:10.7717/peerj.17398_

## Round 0.1 · original submission · Major Revisions

Please revise the manuscript based on the reviewers' comments. A point-by-point response is needed for all the questions.

Reviewer 1 ·

Basic reporting

The article was written in clear and unambiguous English, it conforms to the professional standards of expression, while the format of some words need to be revised to be more professional:
1. Throughout the article, some places use “autoVSA”, some places use “AutoVSA”, like in line 111, 212 and 217. I’d suggest unifying it.
2. Please unify the format of the headlines. Like line 103, 123 and 211, the letters of some words are not capitalized.
3. Typo in line 128, maybe consider “codes” to “code”?
4. Line 187, maybe consider replacing “Yolact” with “YOLACT”.
5. The explanation of “mAP” would be better to be at the first appearance (line 181) instead of at line 202.

Literature references of most citations are provided. But some places are missing citations or not clear:
1. Citations to YOLACT and DeepSort are missing in line 134.
2. The explanation and citation of the “standard test dataset” in line 179 need to be provided.

The overall structure of the article follows the standard of a research article. But it seems that line 185 to line 189 is discussing the results, maybe it would be better to be moved to the Result section?

The article is self-contained with relevant results to the main objective.

Experimental design

The topic of this article is about using deep learning techniques to automate the identification and segmentation of the urine spots, which is within the aims and scope of PeerJ.

The research question is about how to automate the identification and segmentation of the urine spots, which is well stated and meaningful.

Authors investigated the research question with a high technical standard.

The methods used in the experiments are overall clearly described with some places that can be elaborated:
1. The formula that the authors used to calculate the AP/mAP (line 181) is not provided, which may lead to some ambiguity.
2. The authors mentioned small, medium and large scales in line 188, but the precise definitions of small, medium and large are not provided.
3. In Table1, it might be more helpful for readers from medical or biological areas to explain the meaning of AP50 and AP75. Like AP50 means the average precision using IoU of 50% as the threshold.

Validity of the findings

In line 205, the authors claimed that YOLACT has “low omission rate”, but in the article, only precision metrics are provided. It would be better to provide some recall metrics to support the “low omission rate” conclusion. I looked into Figure 4 Mouse 1 and Mouse 3, it seems Solov2 identified more urine spots than YOLACT.

In Table1, only two methods are compared with each other. I’d suggest adding more comparisons like YOLOv7 to make the conclusion “YOLACT segmentation outperforms other methods, demonstrating high accuracy and low omission rates in detecting and segmenting urine spots” (line 204 - 205) more solid.

In addition, some discussions about model accuracy compared to human accuracy could make the article more comprehensive. On the other side of the coin, authors could discuss how the model prediction error could influence the analysis of urination behavior and the study of the neural mechanism underlying urination when evaluating the impact of the deep learning-based method.

Reviewer 2 ·

Basic reporting

All comments have been added in detail to the 4th section called additional comments.

Experimental design

All comments have been added in detail to the 4th section called additional comments.

Validity of the findings

All comments have been added in detail to the 4th section called additional comments.

Additional comments

Review Report for PeerJ Computer Science
(Automated identification and segmentation of urine spots based on deep-learning)

1. Within the scope of the study, segmentation of urine spot images of mice was performed using deep learning.

2. The use of urine spot mouse images collected specifically for the study from the Laboratory Animal Center at the Guangxi University, instead of open source datasets, shows the originality of the study in terms of its dataset.

3. The literature review is not done in-depth in the Introduction section. In addition, it is recommended that the difference of the study from the literature and its main contributions to the literature be added in more clear and precise form.

4. More detailed information about the dataset used within the scope of the study should be given in the Materials & Methods section. Information such as total dataset amount, classes, image resolution, data augmentation/preprocessing steps, training, validation and test dataset amount and percentages can be added in more detail and in tables, etc., if available.

5. Explain in detail how the dataset distribution was determined in the Training of autoVSA section, whether different experiments were made, and how hyperparameter values such as learning rate and optimizer were selected.

6. Explain in detail how the deep learning models used for the segmentation of urine spot images were determined within the scope of the study, have different models been tried, and why these models were preferred, although there are many different models that can be used for segmentation in the literature.

7. When the study is examined in terms of deep learning models used in segmentation, it is very difficult to talk about originality in terms of method since both a limited number of existing models are used.

8. ResNet was used as the backbone for the models used in Table-1. Explain in detail why this is preferred. Why were different backbones not used?

9. Evaluation metrics required for accurate analysis of segmentation results must be obtained completely.

10. Although the results obtained depend on the dataset used, it is beneficial to use a certain number of deep learning models for accurate analysis. From this perspective, it may be more appropriate to support the results obtained with a small number of models used within the scope of the study by adding different models or to compare them as a table with different studies in the literature.

As a result, although the study is at a certain level in terms of both the problem addressed and the dataset used, explaining and/or performing the steps mentioned above will further increase the quality of the study and its contribution to the literature.

Reviewer 3 ·

Basic reporting

This study uses deep learning methods to explore the detection and segmentation of urine spots, which is quite interesting. There are several issues:
1. The standardization of reference format needs to be further improved;

Experimental design

The overall design is reasonable, but there are several doubts:
1. What is the purpose of inferring urine volume from urine spots;

Validity of the findings

1. The importance of using the watered algorithm for identifying overlapping urine spots has been repeatedly mentioned in the article, but no corresponding content has been seen in subsequent ablation experiments, making the conclusion somewhat hollow.
2. The ablation experiment is not entirely complete, for example, only discussing the case where the backbone network is Resnet50.
3. It is recommended to provide original data, code, and website, otherwise some content cannot be verified; For example, What is the content of manual labeling?

Additional comments

None

Reviewer 4 ·

Basic reporting

no comment

Experimental design

no comment

Validity of the findings

no comment

Annotated reviews are not available for download in order to protect the identity of reviewers who chose to remain anonymous.

---

## Round 0.2 · Minor Revisions

Dear authors, please revise the manuscript by addressing the final remaining minor comments of the reviewers

Reviewer 1 ·

Basic reporting

My comments and questions on the "Basic reporting" in the first round of review have been resolved.

Experimental design

My comments and questions on the "Experimental design" in the first round of review have been resolved.

Validity of the findings

Most of my comments and questions on the "Validity of the findings" in the first round of review have been resolved. The remaining one is "In Table1, only two methods are compared with each other. I’d suggest adding more comparisons like YOLOv7 to make the conclusion “YOLACT segmentation outperforms other methods, demonstrating high accuracy and low omission rates in detecting and segmenting urine spots” (line 204 - 205) more solid." The authors have properly responded to this comment, however, due to resource limit, they can only compare two models now. I would suggest the authors to include more models in comparison in the future to make the conclusion more solid.

Reviewer 2 ·

Basic reporting

All comments have been added in detail to the 4th section called additional comments.

Experimental design

All comments have been added in detail to the 4th section called additional comments.

Validity of the findings

All comments have been added in detail to the 4th section called additional comments.

Additional comments

Review Report for PeerJ Computer Science
(Automated identification and segmentation of urine spots based on deep-learning)

Thanks for the revision. I examined the changes made to the paper and the responses to the reviewer comments in great detail. Although some of the answers given are limited, I recommend that this paper be accepted due to its contribution to the literature and its originality at a certain level. I wish the authors success in their new study. Kind regards.

Reviewer 4 ·

Basic reporting

no comment

Experimental design

no comment

Validity of the findings

no comment

Annotated reviews are not available for download in order to protect the identity of reviewers who chose to remain anonymous.

---

## Round 0.3 · accepted · Accept

The authors have addressed the reviewer's concerns and comments well, and the manuscript is in good form for publication now.